# Head-to-Head Comparison: P-POSSUM and ACS-NSQIP^®^ in Predicting Perioperative Risk in Robotic Surgery for Gynaecological Cancers

**DOI:** 10.3390/cancers16132297

**Published:** 2024-06-22

**Authors:** Lusine Sevinyan, Hasanthi Asaalaarchchi, Anil Tailor, Peter Williams, Matthew Evans, Darragh Hodnett, Darshana Arakkal, Pradeep Prabhu, Melanie S. Flint, Thumuluru Kavitha Madhuri

**Affiliations:** 1Department of Gynaecological Oncology, Royal Surrey NHS Foundation Trust, Guildford GU2 7XX, UKdocmadhuri231@doctors.org.uk (T.K.M.); 2School of Applied Sciences, University of Brighton, Brighton BN2 4GJ, UK; 3Department of Maths and Statistics, University of Surrey, Guildford GU2 7XH, UK; 4Department of Anaesthetics, Royal Surrey NHS Foundation Trust, Guildford GU2 7XX, UKpradeep.prabhu@nhs.net (P.P.)

**Keywords:** risk prediction, preoperative, morbidity, mortality, gynaecological oncology, P-POSSUM, ACS-NSQIP

## Abstract

**Simple Summary:**

Estimating the risk of postoperative complications is important for shared decision-making with the patient. This also helps multidisciplinary teams to plan preoperative management and postoperative care and predict duration of recovery. In this retrospective pilot study, we aim to evaluate the accuracy and reliability of the P-POSSUM and ACS-NSQIP surgical risk calculators in predicting postoperative complications in gynaecological–oncological (GO) robotic surgery (RS). We were able to demonstrate that the ACS-NSQIP risk tool showed 90% accuracy in prediction of five major complications and mortality.

**Abstract:**

*Purpose*: In this retrospective pilot study, we aim to evaluate the accuracy and reliability of the P-POSSUM and ACS-NSQIP surgical risk calculators in predicting postoperative complications in gynaecological–oncological (GO) robotic surgery (RS). *Methods*: Retrospective data collection undertaken through a dedicated GO database and patient notes at a tertiary referral cancer centre. Following data lock with the actual post-op event/complication, the risk calculators were used to measure predictive scores for each patient. Baseline analysis of 153 patients, based on statistician advice, was undertaken to evaluate P-POSSUM and ACS-NSQIP validity and relevance in GO patients undergoing RS performed. *Results*: P-POSSUM reports on mortality and morbidity only; ACS-NSQIP reports some individual complications as well. ACS-NSQIP risk prediction was most accurate for venous thromboembolism (VTE) (area under the curve (AUC)-0.793) and pneumonia (AUC-0.657) and it showed 90% accuracy in prediction of five major complications (Brier score 0.01). Morbidity was much better predicted by ACS-NSQIP than by P-POSSUM (AUC-0.608 vs. AUC-0.551) with the same result in mortality prediction (Brier score 0.0000). Moreover, a statistically significant overestimation of morbidity has been shown by the P-POSSUM calculator (*p* = 0.018). *Conclusions*: Despite the limitations of this pilot study, the ACS-NSQIP risk calculator appears to be a better predictor of major complications and mortality, making it suitable for use by GO surgeons as an informed consent tool. Larger data collection and analyses are ongoing to validate this further.

## 1. Introduction

Evidence suggests that despite undergoing a preoperative consultation and participating in the informed consent process, patients frequently exhibit limited understanding regarding potential postoperative complications [1,2,3]. There is cumulative evidence in the literature to suggest that decision aids often increase patients’ understanding of proposed treatment and interventions and assist in the informed consent process [4].

Hence, estimating the risk of postoperative complications is crucial in the shared decision-making process. It helps clinicians and multidisciplinary teams to plan preoperative management and postoperative care, including inpatient stay, and to predict likely duration of recovery.

Over the last decade, the burden of cancer incidence as well as mortality has grown rapidly worldwide and is reflected in the rates of gynaecological cancer increase, with cervical and uterine cancer incidence and mortality being the seventh most common cancers in women [5]. With the increasing rates of GO operations, especially technically challenging minimally invasive surgery (MIS) [6,7] in obese patients and patients with multiple comorbidities, it is even more important to be able to accurately predict the likelihood of perioperative complications and involve patients in joint decision-making about their surgery. Iyer et al. were able to demonstrate, in their prospective multi-centre study, that the intraoperative rate of complications amongst patients undergoing surgery for cancer was 5.4%, whereas the postoperative complications rate was 27.1% [8].

According to a recent review, cardiopulmonary exercise testing (CPET) is providing an objective evaluation of cardiorespiratory fitness and functional capacity resulting in individualised risk profiles, which could guide shared decision-making [9]. However, although it is increasingly being used in the UK to evaluate preoperatively the postoperative morbidity, it is a very expensive method with limited capacity [10].

Several online risk scoring tools are in clinical use, all of which attempt to predict the risk of postoperative morbidity and mortality during major surgical procedures. One such calculator, originally developed by Copeland et al. in 1991, is the Physiologic and Operative Severity Score for the Enumeration of Mortality and Morbidity (POSSUM) risk scoring [11], subsequently modified to Portsmouth POSSUM (P-POSSUM) to provide a more accurate prediction [12], which is widely accepted in the UK for postoperative mortality and morbidity risk prediction. P-POSSUM uses the preoperative physiological scores and intraoperative surgical scores of patients for generation of postoperative risk scores [11]. However, published data across different specialties, including GO, suggest that it may overestimate risk, especially the mortality rates [13,14,15,16,17], which may cause undue patient anxiety, influence the surgical decision-making and potentially prolong the length of stay (LOS). Moreover, P-POSSUM requires intraoperative parameters for morbidity and mortality calculation, which in turn makes the prediction less accurate.

The American College of Surgeons National Surgical Quality Improvement Program (ACS-NSQIP) surgical risk calculator was developed in 2013 based on the NSQIP database, which was designed to measure and improve the quality of surgical care [18]. The database was derived from standardised prospective surgical data collection, across more than 700 hospitals and nine specialties in the USA, and further evaluation of the postoperative complications. The calculator derived from this database is a validated web-based tool requiring 19 preoperative risk factors for the prediction of LOS and 11 postoperative outcomes within 30 days of surgery (plus 2 outcomes in surgery involving gastrointestinal tract—ileus and anastomotic leak), which allows surgeons to receive a customised, patient-specific risk for their surgery [19].

A number of studies have explored the applicability and efficacy of the ACS-NSQIP surgical risk calculator in the context of GO [20,21,22,23]. However, none of these studies have compared ACS-NSQIP to any other tools and there have not previously been attempts to analyse the tool in a cohort of patients undergoing RS.

The objective of our study was to explore the validity and suitability, as well as to compare the performance, of the P-POSSUM and ACS-NSQIP surgical risk calculators in the setting of GO minimally invasive robotic-assisted operations and hereby we present our pilot study that shows the performance of ACS-NSQIP compared to P-POSSUM.

## 2. Ethics and Approvals

This project was approved by the Audit and Quality Improvement department as part of the Research and Development office of the Royal Surrey NHS Foundation Trust (RSNFT) as part of the studentship with University of Brighton for a retrospective analysis of 1500 patients undergoing RS and 1500 patients undergoing open surgery.

## 3. Materials and Methods

### 3.1. Cohort Selection

Yearly operation lists were obtained from the designated electronic GO database of RSNFT. All patients were divided by the year of the operation (2009–2020) and by the type of surgery they had undergone (open/robotic/laparoscopic). Women who underwent other diagnostic procedures have been excluded from this study, as well as cases where there was not enough information for the ACS-NSQIP score calculation or the P-POSSUM score. While data collection is ongoing, the first 153 robotic cases were felt to be adequate for statistical analysis for this pilot and hence the first 153 robotic cases with complete data have been analysed.

In this report, we present the preliminary analysis of 153 patients who had undergone Da Vinci-assisted RS for suspected or confirmed gynaecological malignancies at RSNFT, based on the intention to treat analysis.

### 3.2. Data Collection

For all patients undergoing surgery at our cancer centre, data are recorded prospectively on a dedicated database. For each patient, the electronic database record was retrieved to collect P-POSSUM scores, along with the 19 pre-operative parameters required to calculate the ACS-NSQIP risk (Table 1). The missing data were collected from paper patient records.

Following their surgery and discharge, patients have the option of contacting their local hospital, general practitioner (GP) or RSNFT for any concerns, and all patients at RSNFT are invited for a follow-up appointment within 30 days postoperatively when the course of recovery +/− further management/surveillance is discussed with the treating specialist. All readmissions and presentations to different trusts, visits to GP and treatments commenced within that period are discussed with the patient. Clinical nurse specialists are also contacted by other trusts when patients present/are admitted to other hospitals. These events are documented within the electronic database and/or patients’ records. The electronic database and paper records were also analysed to identify any actual postoperative 30-day complications and the LOS, which were recorded along with the rest of the data. Morbidity was defined by either the Clavien–Dindo score and/or the Postoperative Morbidity Survey (POMS) score within the 30 days after the surgery. All complications above Clavien–Dindo IIIa were classified as serious complications.

In addition to the variables listed above, a number of tumour characteristics were collected for each of the patients for general purposes, such as tumour site, neoplasm type and subtype, FIGO stage and grade.

The required parameters were entered into the online ACS-NSQIP surgical risk calculator tool [24] for each patient, and the Common Procedural Terminology (CPT) code was selected as accurately as possible. In accordance with previous studies, if there was more than one procedure within the surgery, then more than one CPT code was selected and the scores for each were calculated; the highest percentage of each of the complications was selected for analysis [20,23]. Finally, the estimated LOS and risk of 11 postoperative outcomes were calculated (Table 2): serious complication/any complication; (1) pneumonia; (2) cardiac complication; (3) surgical site infection (SSI); (4) urinary tract infection (UTI); (5) venous thromboembolism (VTE); (6) renal failure; (7) readmission; (8) return to the theatre; (9) death; (10) discharge to post-acute care; (11) sepsis.

### 3.3. Statistical Analysis

Firstly, baseline demographic variables were analysed to describe the selected cohort. Patients were categorised by age, ASA class, degree of obesity and types of comorbidities. Tumours were categorised as being malignant or benign, as well as by the site of neoplasm.

Statistical analysis was undertaken using Statistical Package for the Social Sciences (SPSS) version 24.0. The difference between each of the predicted postoperative outcomes, hence the definitiveness of the ACS-NSQIP surgical risk calculator, and the actual complications were measured using the Brier score. The closer the reliability term was to zero, the more reliable was the prediction of morbidity, mortality by P-POSSUM and ACS tools or specific complication by the ACS tool (https://riskcalculator.facs.org/RiskCalculator/PatientInfo.jsp, accessed 2 March 2023).

Within the logistic regression model, we also determined the c-statistic standard measure of the predictive accuracy, which is equivalent to the area under the Receiver Operating Characteristic (ROC) curve when the outcome is binary [25]. The ROC curve values range from 0 to 1, where 0 indicates an ineffective model and 1 means that the model is perfect for the prediction of complications [26].

## 4. Results

In this report, we present results for the initial 153 patients. These patients had undergone a Da Vinci RS for gynaecological malignancy at RSNFT. Preoperative demographic data are presented in Table 3. The vast majority of the patients were under 65 years of age (56.2%) and were classed as ASA 2 (66%). All patients had undergone planned procedures with no emergencies. Similar numbers of patients were of normal weight and overweight (30% and 27%, accordingly) and the majority of the patients were classed as obese (class 1–3) (43%).

In terms of comorbidities, a substantial proportion of the patients were found to have hypertension (35.3%), followed by 15% of smokers and 9.8% of non-insulin-dependent diabetes mellitus patients. Only a small number of patients were suffering from insulin-dependent diabetes mellitus (2.6%) and COPD (0.6%).

The majority of patients underwent surgery for malignancy (85%); 4.6% had borderline condition and 10.4% were found to have a benign tumour. A total of 64.7% of the tumours were located in endometrium, 29.4% in the cervix, 5.2% in the ovaries and one patient had a primary in another site (Table 4).

The main cohort of patients had undergone hysterectomy with or without salpingoophorectomy and a staging procedure, lymphadenectomy and/or omentectomy with or without hysterectomy and/or salpingoophorectomy. The breakdown of the patients by the type of surgery is shown in Table 4.

Of the 153 patients, 23 (15%) were found to have one or more complications and 13 (8.5%) had an event constituting a serious complication, as described previously. The breakdown of the actual postoperative complications is demonstrated in Table 5. As shown, the most frequent complications in the sequence of their frequency are UTI (*n* = 10), readmission (*n* = 7), pneumonia (*n* = 3) and SSI (*n* = 3). Rarer complications included return to theatre (*n* = 2), VTE (*n* = 2), renal failure (*n* = 1) and discharge to post-acute care facility (*n* = 2). There were no deaths or cardiac complications observed in this cohort of patients.

When comparing the morbidity prediction ability of P-POSSUM and ACS-NSQIP surgical risk calculators, ACS-NSQIP showed a better predictive value than P-POSSUM with AUC 0.608 against 0.551 (Figure 1). ACS-NSQIP was also found to statistically significantly predict postoperative complications better than P-POSSUM in patients with ASA class I and II.

The ACS-NSQIP calculator was found to perform best in predicting the risk of postoperative VTE (AUC = 0.793) and pneumonia (AUC = 0.657), followed by a worse prediction of readmission (AUC = 0.587) and UTI (AUC = 0.515). (Figure 2).

When comparing the mortality predictions by ACS-NSQIP and P-POSSUM, the best results were shown by ACS-NSQIP (Brier (P-POSSUM mortality) = 0.0014; Brier (ACS mortality) = 0.000; *p* = 0.084) (Figure 3a) (Table 6).

Given the lack of deaths in the cohort, it is evident that the P-POSSUM risk calculator is overestimating the mortality prediction compared to the ACS risk calculator, which has accurately predicted the mortality.

P-POSSUM was also found to statistically significantly overestimate morbidity, similarly to mortality risks (*p* = 0.018), as shown on Figure 3b and Table 6.

Lastly, the performance of ACS-NSQIP in predicting individual complications was measured using the Brier score. The risk prediction tool showed the best accuracy in the prediction of major complications as follows: death, cardiac event, renal failure, VTE, return to theatre and SSI (Figure 4) (Table 7).

## 5. Discussion

To our knowledge, this study has been the first attempt to validate and compare the predictive value of surgical risk calculators, P-POSSUM and ACS-NSQIP, in the context of RS in GO. Previous studies have identified the lack of accuracy in the risk prediction of the P-POSSUM tool. It has been shown that P-POSSUM tends to overestimate mortality risk prediction in patients undergoing different types of surgery, specifically in GO [13,27], and our study has also confirmed that. This can affect the potential patients’ understanding of possible outcomes of surgery and influence the recovery, as the psychological aspect plays a vital role in cancer patient recovery [28]. Moreover, we have been able to show that P-POSSUM overestimates morbidity along with mortality scores amongst our cohort of patients, which makes it an unreliable and potentially harmful tool to be used in this group of patients.

On the other hand, the results of our study show that ACS-NSQIP would have higher predictive value and would be a better tool to be used in GO. A study by Murray et al. with 142 patients had shown that ACS-NSQIP has a potential applicability in GO, especially since it could be used as a visual aid when counselling patients before the operation [22]. ACS-NSQIP has been shown to be an accurate predictor, especially for death and VTE, which is in agreement with the study by Szender et al., which demonstrated that the Brier score approached the threshold for death, pneumonia, UTI and VTE in cancer patients, although the patient cohort of *n* = 577 appear to have all had open surgery [29]. Rivard et al. in their cohort of 1094 open cases were able to demonstrate a c-statistic value of >0.8 for death and 0.7 for cardiac events and renal failure [20].

Thigpen et al. in their retrospective study, which included a cohort of patients who had undergone total abdominal or laparoscopic hysterectomy for benign gynaecological disease, found that the ACS-NSQIP calculator was effective in prediction of VTE and SSI but did not perform well in other categories of complications [30]. These previous studies seem to be in concordance with our current research with regards to the efficacy of preoperative prediction of VTE during the postoperative period. In contrast, Teoh et al. report that the ACS-NSQIP risk calculator performs poorly in predicting 30-day postoperative complications in GO patients, especially compared to the general and colorectal surgical population [21]. In their study, a mixed population of patients who had undergone MIS (laparoscopic or robotic) was reviewed in contrast to our pilot study, where we have reviewed exclusively robotic cases. This could influence the result, as patients selected for RS cases often have multiple comorbidities and are considered more complex cases. Manning-Geist et al. in their study (*n* = 261) had also shown that ACS-NSQIP performed poorly in patients with ovarian cancer undergoing interval debulking surgery [23].

Past studies have also shown that the presence of the following parameters increases the risk of postoperative complications in GO operations: diabetes, any comorbidity, increasing ASA class, age and BMI. The risk also depended on whether the approach was laparoscopic or open [8]. These listed parameters are all included in the ACS-NSQIP surgical risk calculator. Nonetheless, Iyer et al. also showed that intra-operative factors also play a significant role in the prediction of postoperative complications. Those include estimated blood loss, duration of surgery and previous abdominal surgery. [8] This could potentially be the reason the ACS-NSQIP tool has not shown such accurate predictive abilities in GO patients compared to the colorectal surgical cohort [31]. Other reasons, as described by Rivard et al., could include the variety of complexity of procedures because of intraperitoneal disease burden, the need for a greater amount of CPT codes to be taken into account considering the complexity of the procedures and the frequent poor nutritional status of the GO patients weeks prior to the operations [20].

## 6. Limitations and Insights

Being a pilot, retrospective data collection, the authors acknowledge that this poses a limitation to our study, potentially resulting in underreporting of postoperative complications due to inadequate follow-up or insufficient documentation in medical records.

All attempts were made to obtain complete records to obtain the results from both the ACS and P-POSSUM calculators for comparison. Nonetheless, data collection is currently underway with a much larger cohort of patients undergoing both open and minimally invasive surgery as well as validation in a prospective setting. It has been difficult to account for rare postoperative events, considering the number of participants, but plans exist to overcome this limitation in the future. The demographic profile of Surrey reveals a population characterised by relatively favourable ageing and affluence in comparison to broader demographics observed across the United Kingdom, though obesity as a risk factor prevails. In the ongoing data analysis, we strategically integrate multi-centre datasets to mitigate potential biases from specific demographic profiles. Lack of case selection has resulted in cases across all stages being incorporated, which the authors acknowledge is a limitation as there is evidence that the complexity of surgery is related to the disease burden as well as patient factors. Once again, being a pilot study, it does not allow for a stratified data analysis, which will be possible with larger datasets in the future.

This is the first study to analyse a separate cohort of patients who underwent Da Vinci robot-assisted operations, which has not been undertaken previously. Furthermore, a head-to-head comparison of the performance of two surgical risk calculators, P-POSSUM and ACS-NSQIP, in this cohort of patients, which is the main strength of this study, and has been undertaken at a tertiary referral cancer centre where patients are referred following a suspected or confirmed case of malignancy.

This preliminary analysis suggests a notable disparity in the prognostic accuracy between the ACS-NSQIP scoring system and the P-POSSUM model in the context of mortality prediction, discerning a distinct advantage in favour of the ACS-NSQIP scoring system, which demonstrated a null mortality rate, as opposed to the P-POSSUM model, which failed to anticipate such outcomes. This observation bears significant implications for our ongoing research endeavours, substantiating the heightened efficacy of the ACS-NSQIP scoring system in mortality prediction and thereby providing a more robust foundation for future investigations.

## 7. Conclusions

This pilot study, along with others, suggests that the P-POSSUM surgical risk tool overestimates morbidity and mortality scoring in GO patients undergoing RS and as such should not be the preferred risk calculator for this cohort of patients. The ACS-NSQIP surgical risk calculator informative value has been higher, especially in predicting major complications, and it may be used as an informed consent tool. However, future larger studies are necessary to further evaluate these prediction tools.

## Figures and Tables

**Figure 1 cancers-16-02297-f001:**
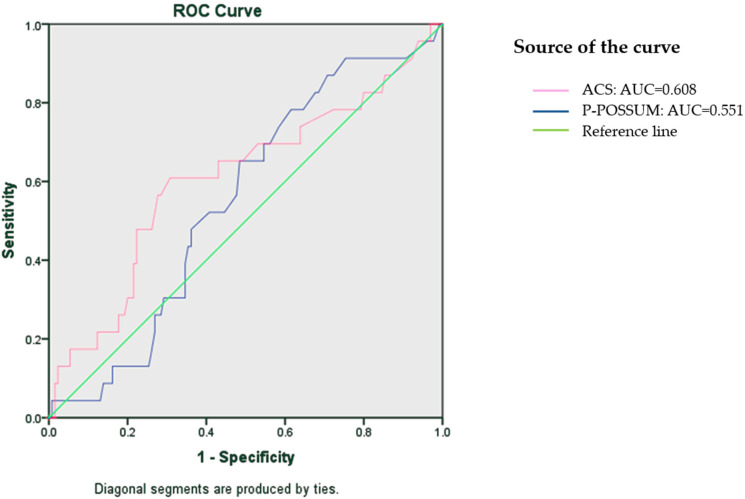
The ROC area under the curve (AUC), which ranges from 0.0 to 1.0, is a diagnostic measure for evaluating the accuracy of a predictor. The diagonal green reference line has an AUC of 0.5 and means random guessing. This ROC curve represents the comparative analysis of ACS-NSQIP and P-POSSUM general morbidity prediction.

**Figure 2 cancers-16-02297-f002:**
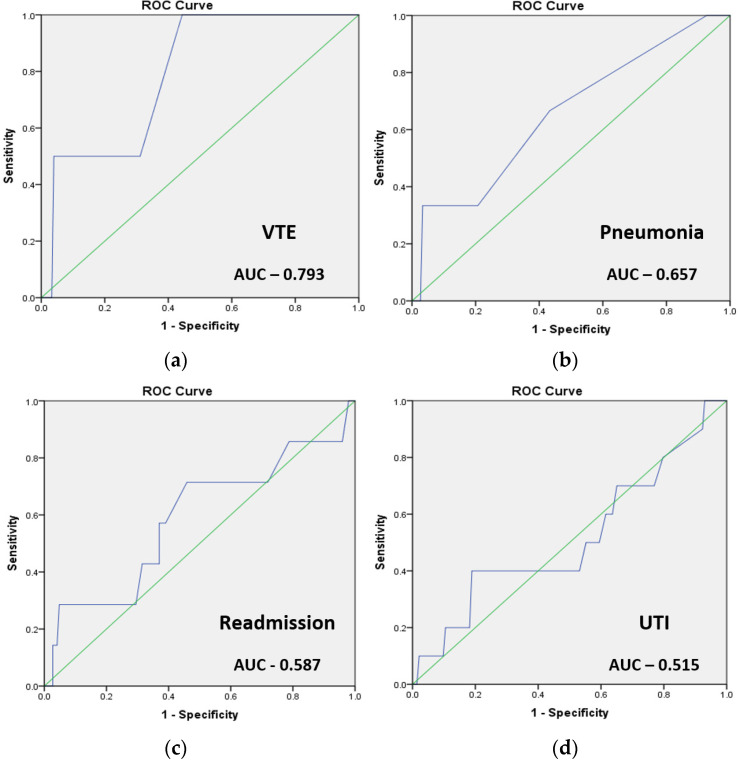
Logistic regression charts for postoperative prediction value of the ACS-NSQIP surgical risk calculator. (**a**) VTE (AUC = 0.793), (**b**) pneumonia (AUC = 0.657), (**c**) readmission (AUC = 0.587), (**d**) UTI (AUC = 0.515). The diagonal green reference line has an AUC of 0.5 and means random guessing. This ROC curves represent the analysis of specific complications ACS-NSQIP predictions.

**Figure 3 cancers-16-02297-f003:**
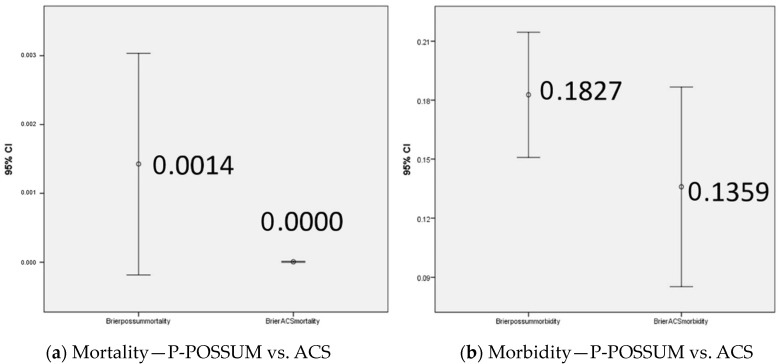
Brier score measures the accuracy of probabilistic predictions: the lower the Brier score is for a set of predictions, the better the predictions are calibrated. (**a**) Mortality prediction: ACS (Brier Score 0.000), P-POSSUM (Brier score 0.0014). (**b**) Morbidity prediction: ACS (Brier score 0.1827), P-POSSUM (Brier score 0.1359).

**Figure 4 cancers-16-02297-f004:**
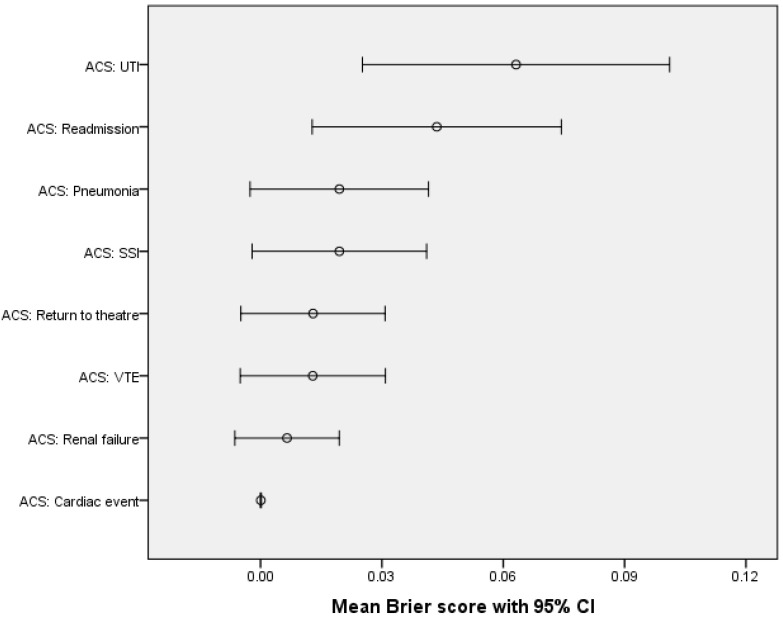
ACS-NSQIP predictive ability of postoperative complications.

**Table 1 cancers-16-02297-t001:** ACS-NSQIP surgical risk calculator preoperative parameters.

Preoperative Parameters	Patient-Specific Options
Age group	Under 65 years/65–74 years/75–84 years/85 years or older
Sex	Female
Functional status	Independent/Partially Dependent/Totally Dependent
Emergency case	No/Yes
ASA class	I (healthy patient)/II (mild systemic disease)/III (severe systemic disease)/IV (severe systemic disease/constant threat to life)/V (moribund/not expected to survive surgery)
Steroid use for chronic condition	No/Yes
Ascites within 30 days of surgery	No/Yes
Systemic sepsis within 48 h prior to surgery	None/SIRS/Sepsis/Septic shock
Ventilator dependent	No/Yes
Disseminated cancer	No/Yes
Diabetes	No/Yes (on oral medication)/Yes (on insulin)
Hypertension requiring medication	No/Yes
Congestive heart failure in 30 days prior to surgery	No/Yes
Dyspnoea	No/With moderate exertion/At rest
Current smoker within 1 year	No/Yes
History of severe COPD	No/Yes
Dialysis	No/Yes
Acute renal failure	No/Yes
BMI	Kg/m^2^

**Table 2 cancers-16-02297-t002:** ACS-NSQIP surgical risk calculator postoperative outcomes.

Outcome	Inclusion Criteria
Serious complication	Cardiac arrest, myocardial infarction, pneumonia, progressive renal insufficiency, acute renal failure, PE, DVT, return to the theatre, deep incisional SSI, organ space SSI, systemic sepsis, unplanned intubation, UTI, wound disruption.
Any complication	Superficial incisional SSI, deep incisional SSI, organ space SSI, wound disruption, pneumonia, unplanned intubation, PE, DVT, ventilator > 48 h, progressive renal insufficiency, acute renal failure, UTI, stroke, cardiac arrest, myocardial infarction, return to the theatre, systemic sepsis.
Pneumonia	Diagnosed using both radiologic (i.e., infiltrate, consolidation or opacity, cavitation) and clinical (e.g., fever, leukopenia/leukocytosis, culture results, patient symptoms) criteria.
Cardiac complication	Cardiac arrest or myocardial infarction.
SSI	Superficial incisional SSI, deep incisional SSI or organ space SSI.
UTI	Diagnosed using a combination of clinical symptoms and laboratory confirmation (e.g., urine culture, pyuria, positive dipstick) or initiation of appropriate antimicrobial therapy.
VTE	New thrombus within the venous system requiring therapy.
Renal failure	Progressive renal insufficiency or acute renal failure requiring dialysis.
Readmission	-
Return to theatre	Return to theatre for additional surgery that was not planned at the time of the initial surgery.
Death	-
Discharge to post-acute care	Discharge to a nursing home or rehabilitation facility.
Sepsis	-
Ileus *	Prolonged Postoperative NPO or NGT Use: Prolonged NPO status or NGT use for suctioning or decompression, more than 3 days postop (POD4 or later) OR reinsertion of NGT or reinstating NPO status any time POD4 or later within 30 days.
Anastomotic leak *	This includes air, fluid, GI contents or contrast material. With or without treatment. The presence of an infection/abscess thought to be related to an anastomosis, even if the leak cannot be definitively identified as visualised during an operation, or by contrast extravasation, would still be considered an anastomotic leak if this is indicated by the surgeon.

* For operations involving GI tract.

**Table 3 cancers-16-02297-t003:** Preoperative demographic data.

Variable	Overall Patients *n* = 153 N	%
** Age Group **		
Under 65 years	86	56.2%
65–74 years	38	24.8%
75–84 years	24	15.7%
85 years or older	5	3.3%
*Mean age*	*60*	
*Standard deviation*	*12.1*	
** Functional Status **		
Independent	152	99.4%
Partially dependent	1	0.6%
Totally dependent	0	0%
** ASA Class **		
I	22	14.4%
II	101	66.0%
III	30	19.6%
IV	0	0%
V	0	0%
** Steroid Use **		
No	150	98.0%
Yes	3	2.0%
** Ascites within 30 Days **		
No	153	100.0%
Yes	0	0%
** Disseminated Cancer **		
No	138	90.2%
Yes	15	9.8%
** Diabetes **		
No	134	87.6%
Oral	15	9.8%
Insulin	4	2.6%
** Hypertension **		
No	99	64.7%
Yes	54	35.3%
** Heart Failure **		
No	152	99.4%
Yes	1	0.6%
** Dyspnoea **		
No	130	85.0%
With moderate exertion	23	15.0%
At rest	0	0%
** Smoker **		
No	130	85.0%
Yes	23	15.0%
** Severe COPD **		
No	152	99.4%
Yes	1	0.6%
** BMI **		
Underweight (<18.5)	0	0%
Normal (18.5–24.9)	46	30.0%
Overweight (25.0–29.9)	41	26.8%
Obese Class 1 (30.0–34.9)	28	18.3%
Obese Class 2 (35.0–39.9)	14	9.2%
Obese Class 3 (>39.9)	24	15.7%
*Mean BMI*	*29.6*	
*Standard Deviation*	*8.63*	

**Table 4 cancers-16-02297-t004:** Breakdown of the patients by tumour site, staging and the type of operation undergone.

Tumour Site	N	%		N	FIGO Stage	N
Ovary	8	5.2	Benign	4	n/a	n/a
			Borderline	1	1a	1
			Invasive	3	1c	1
					4a	1
					4b	1
Ovary operations	Uni-/bilateral salpingo- (and/or) oophorectomy					8
	Total laparoscopic hysterectomy					6
	Peritoneal washing					3
	Adhesiolysis					2
	Supracolic omentectomy					2
	Modified radical hysterectomy					1
	PLNS					1
	PALNS					1
	Infracolic omentectomy					1
	Appendicectomy					1
	Omental biopsy					1
Uterus	99	64.7	Benign	11	n/a	n/a
			EIN	5	n/a	n/a
			Invasive	83	1a	43
					1b	25
					2	4
					2b	1
					3a	1
					3b	2
					3c1	2
					3c2	1
					4a	1
					4b	3
Uterus operations	Total laparoscopic hysterectomy					98
	Uni-/bilateral salpingo- (and/or) oophorectomy					94
	Peritoneal washing					80
	PLNS					43
	Omental biopsy					16
	PALNS					15
	Adhesiolysis					9
	Mini-laparotomy					7
	Bilateral sentinel node assessment					3
	Excision of nodules from peritoneum					2
	Peritoneal biopsy					2
	Modified radical hysterectomy					1
	Infracolic omentectomy					1
	Appendicectomy					1
	Biopsy of lesion on round ligament					1
	Laparoscopic myomectomy					1
	Repair of intra-operative bladder injury					1
	Suture of vaginal tear					1
Cervix	45	29.4	CIN	1	n/a	n/a
			Invasive	44	1a	6
					1b	33
					2a	1 *
					2b	2 *
Cervix operations	Uni-/bilateral salpingo- (and/or) oophorectomy					242
	PLND					23
	Cystoscopy					18
	Laparoscopic radical hysterectomy					16
	Rigid sigmoidoscopy					10
	Total laparoscopic hysterectomy					7
	Insertion of cervical cerclage					3
	Sentinel node assessment					2
	Radical trachelectomy					2
	Vaginal trachelectomy					2
	Ovarian transposition					1
	Radical upper vaginectomy					1
	PLNS					1
	PALNS					1
	Zoladex injection					1
	Drainage of lymphocyst					1
	Excision of vulval lesion					1
	Ureteral stenting					1
	Peritoneal washings					1
Other	1	0.7	n/a	n/a	n/a	n/a
Other operations	TLH, BSO, supracolic omentectomy					1

* Patient underwent two-stage procedure.

**Table 5 cancers-16-02297-t005:** Rates of complications which occurred within 30 days of the operation.

	Complication(s) Occurring within 30 Days	N	Mean	Std. Deviation	Std. Error Mean
POSSUM morbidity	No	130	30.926	18.319	1.607
	Yes	23	31.869	15.310	3.192
ACS any complications	No	130	4.920	2.524	0.221
	Yes	23	5.865	2.955	0.616
ACS pneumonia	No	150	0.212	0.323	0.028
	Yes	3	0.209	0.176	0.037
ACS cardiac	No	153	0.172	0.892	0.078
	Yes	0	0.096	0.149	0.031
ACS SSI	No	150	1.882	0.926	0.081
	Yes	3	2.100	1.009	0.210
ACS UTI	No	143	1.885	1.195	0.105
	Yes	10	2.400	1.426	0.297
ACS VTE	No	151	0.387	0.245	0.022
	Yes	2	0.504	0.348	0.073
ACS RF	No	152	0.080	0.155	0.014
	Yes	1	0.135	0.187	0.039
ACS readmission	No	146	3.425	1.863	0.163
	Yes	7	3.943	1.986	0.414
ACS return to theatre	No	151	1.212	0.394	0.035
	Yes	2	1.183	0.351	0.073

**Table 6 cancers-16-02297-t006:** Paired T-test sample statistics for morbidity and mortality prediction of P-POSSUM and ACS risk calculators.

		Mean	N	Std. Deviation	Std. Error Mean
Pair 1	Brier P-POSSUM morbidity	0.183	153	0.199	0.016
	Brier ACS morbidity	0.136	153	0.318	0.026
Pair 2	Brier P-POSSUM mortality	0.001	153	0.010	0.001
	Brier ACS mortality	0.000	153	0.000	0.000

**Table 7 cancers-16-02297-t007:** Statistics for prediction of individual postoperative complications using the ACS risk calculator.

	N	Mean	Median	Std. Deviation
Brier ACS pneumonia	3	0.020	0.000	0.138
Brier ACS cardiac	0	0.000	0.000	0.001
Brier ACS SSI	3	0.020	0.000	0.135
Brier ACS UTI	10	0.063	0.000	0.238
Brier ACS VTE	2	0.013	0.000	0.112
Brier ACS renal failure	1	0.007	0.000	0.081
Brier ACS readmission	7	0.044	0.001	0.193
Brier ACS return to theatre	2	0.013	0.000	0.112

## Data Availability

Data available upon request.

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
