# Peer review of "Head-to-Head Comparison: P-POSSUM and ACS-NSQIP® in Predicting Perioperative Risk in Robotic Surgery for Gynaecological Cancers"

_cancers, 2024, doi:10.3390/cancers16132297_

Round 1

Reviewer 1 Report (Previous Reviewer 3)

Comments and Suggestions for Authors

I thank the authors for submitting the revisions.

However my original concerns remain - there is no reported mortality and the sample size is too small to assess the accuracy and of the surgical risk calculators. I would recommend resubmitting with larger patient numbers.

Also there have been multiple studies that have attempled to validate the ACS-NSQIP model within the gynae onc surgical setting. All have shown poor performance. This is contrary to this papers findings and not discussed in the discussion

Author Response

Reviewer 2 Report (Previous Reviewer 2)

Comments and Suggestions for Authors

Thanks for the opportunity to review this revised manuscript and to the authors for addressing most of my concerns after the first submission!

I feel that this is close to being ready for submission.  Just a few suggestions:

1. Please further clarify (even in a couple of words or a sentence) why 153 participants?  Were these the first 153 patients with gynecologic malignancy who underwent robotic surgery at your centre ever?  In a fiscal year?  Why 153 still isn't clear to me (why not 140? 160? 211?)

2. Table 6 is can probably be omitted and replaced with text.   You don't need to tabulate a pairwise comparison like this.  Can just write that the pairwise comparisons depicted in Figure 3 were statistically significant (p=  ) using a t test or something like that

3.   If this was truly a pilot study, that should be noted in the abstract, intro, methods, and conclusion; should be clear this was a 'pilot' to assess the logistical feasbility/early findings/etc etc of a larger study that you plan to complete.

Comments on the Quality of English Language

There are still minor grammatical issues that could be corrected.

Author Response

Reviewer 3 Report (New Reviewer)

Comments and Suggestions for Authors

This is a well written retrospective study on the complication analysis in robotic surgery.

The methodology has some limitation regarding the data records, however it add valuable info in this field.

The results are well presented with adequate tables.

It can be published.

Author Response

We are grateful to reviewer 3 for their kind encouragement and support and there has been no suggestion to edit. Thank you.

Reviewer 4 Report (New Reviewer)

Comments and Suggestions for Authors

Dear Authors,

Your article is very interesting and the results are important.

I would suggest to remove that yellow highlights.

Page 4, line 141: the web address should be moved to References

Page 6, Table 3 – you can add sd for the age and BMI as well

Page 9, Table 5 – 3 decimals are enough

Page 11, Figure 2 – I would suggest to redo the graphs by the same version of SPSS as you used for Figure 1

Page 12, Table 6 and Table 7 – 3 decimals are enough

Page 14, line 298 – I would suggest to rewrite the sentence

Author Response

This manuscript is a resubmission of an earlier submission. The following is a list of the peer review reports and author responses from that submission.

Round 1

Reviewer 1 Report

Comments and Suggestions for Authors

In this retrospective analysis, the authors evaluate the accuracy and reliability of P-POSSUM and ACS-NSQIP calculators in predicting postoperative complications in gynecological-oncology robotic surgery (GO-RS). 

P-POSSUM was recently seen as a non-reliable predictor of postoperative morbidity for patients undergoing major gynecological and gastrointestinal surgeries for cancer (Mukherjee S, Kedia A, Goswami J, Chakraborty A. Validity of P-POSSUM in adult cancer surgery (PACS). J Anaesthesiol Clin Pharmacol. 2022 Jan-Mar;38(1):61-65); ACQ-NSQIP was investigated more frequently in both benign and gynecological procedures.

This is the first analysis of comparison between this two different risk assessment model; however there are several limits:

- the aim of the authors is the comparison in the field of oncological gynecology: however 10.4% of the patients included have a benign tumor and 4.6% a borderline form.
These patients should be excluded from the analysis

- in table 4, the authors report ovarian and uterus 4a-4b FIGO stage robotic surgery and 2b and 3a for cervical cancer.
These are patients requiring medical treatment not surgery: why include them in the analysis?

- the prediction of complications depends on the type of surgery performed. In table 5 the procedures are not uniform: there is no mention of type of lymphadenectomy done (aortic and pelvic, or only pelvic) or of type of radicality in hysterectomy; adnexectomy or extra-fascial hysterectomy are routine surgical procedures in GO and they have a different impact than more complex procedures. 

- the total number of patients included in the study (153) is limited and also not specific for GO

- table 3 does not include patients with 0%.

- the ROC curves in figures 2a and 2b are not very specific

- the results and conclusions of the authors confirm what is already known in the literature

Author Response

In this retrospective analysis, the authors evaluate the accuracy and reliability of P- POSSUM and ACS-NSQIP calculators in predicting postoperative complications in gynecological-oncology robotic surgery  (GO-RS).

P-POSSUM was recently seen as a non-reliable predictor of postoperative morbidity for patients undergoing major gynecological and gastrointestinal surgeries for cancer (Mukherjee S, Kedia A, Goswami J, Chakraborty A. Validity of P-POSSUM in adult cancer surgery (PACS). J Anaesthesiol Clin Pharmacol. 2022 Jan-Mar;38(1):61-65); ACQ-NSQIP was investigated more frequently in both benign and gynecological procedures.

We are grateful to the reviewers for advising us of this manuscript which has been on our radar. The authors of the above manuscript have drawn conclusions from analysing 143 patients. However, out of that cohort only 62 patients had undergone a procedure for gynaecological cancer. The authors of this study have also not specified the type of cancer or the outcome of the surgery and the type of procedure that the patients had undergone. Our research has a more homogenic cohort of patients in the type of surgery they underwent and is the first analysis to our knowledge adding to the field of risk prediction in robotic surgery.

As a result, we did not include it in our paper however, since it appeared important that we highlight and make reference to this manuscript, we have done so (line 251 page 14).

This is the first analysis of comparison between this two different risk assessment model; however there are several limits:

  • the aim of the authors is the comparison in the field of oncological gynecology: however 10.4% of the patients included have a benign tumor and 6%  a  borderline  form.  These patients should be excluded  from the analysis.

We appreciate the reviewer feedback and acknowledge this bias. The attempt during this pilot data analysis was to ensure there was no case selection and we have incorporated all the patients consecutively who underwent surgery for a suspected or confirmed gynaecological cancer who were referred to our cancer centre and underwent surgery as per the intention to treat analysis.

We have explained this in our study limitations on page 15-16 (lines 294-327).

  • in table 4, the authors report ovarian and uterus 4a-4b FIGO stage robotic surgery and 2b and 3a for cervical

These are patients requiring medical treatment not surgery: why include them in the analysis?

Thank you for raising an interesting point with regards to the surgical procedures. As mentioned in response to the previous comment, in this pilot, retrospective analysis, case selection has not been undertaken and since the focus of the paper is on risk prediction irrespective of the final diagnosis (benign, borderline or malignant), the authors have included all consecutive cases as per the intention to treat analysis.

However we do thank the reviewer for this feedback and to clarify further, for the patients with ovarian cancer, the preoperative diagnosis following discussion at MDT was not suggestive of an ovarian malignancy and the patients underwent robotic surgery having been fully counselled that in the event ovarian cancer was confirmed, they would require further treatment including a second surgery to fully stage the disease as well as chemotherapy on final histology.

Post the trials, CHORUS by Kehoe et al & the EORTC by Vergote et al, women with no confirmed ovarian cancer such as the patients above have been receiving MIS where feasible to confirm the diagnosis +/- restage if necessary.

Surgery for uterine cancer remains the mainstay of treatment and since the preoperative diagnosis was suggestive of early stage, surgery was considered. However, once the surgery confirmed a more extensive disease, a hysterectomy and BSO was undertaken with no further surgical intervention and hence it is crucial to include these patients in the analyses.

With regards to cervical cancer, we appreciate the reviewers comments and indeed medical management was undertaken first. Following MDT review after completion of all treatment both at 6 weeks and 3 months, imaging and subsequent biopsies suggested residual disease and hence hysterectomy was performed. In two of the patients with advanced disease, bilateral ovarian transposition was the procedure performed prior to medical management. One patient was a node positive 1b1 and was mislabelled as Stage 3a.

  • the prediction of complications depends on the type of surgery In table 5 the procedures are not uniform: there is no mention of type of lymphadenectomy done (aortic and pelvic, or only pelvic) or of type of radicality in hysterectomy; adnexectomy or extra-fascial hysterectomy are routine surgical procedures in GO and they have a different impact than more complex procedures.

We thank the reviewer for raising this point and are entirely in agreement that risk prediction is related to surgical complexity. We have now clarified the types of procedures and incorporated them into Table 4 itself (pages 7-9). It is essential that we recognise that both calculators were originally designed for general surgery per se and neither P-POSSUM nor ACS NSQIP calculators however allow for detailed subclassification of gynaecological oncological procedures. This pilot study has suggested there is a role to explore further using the above calculators and this work is ongoing.

  • the total number of patients included in the study (153) is limited and also not specific for GO

We thank the reviewer for raising this point. This is a pilot study and to this end, a paragraph on study limitations stating that this is a pilot project with only 153 patients from a single site have been evaluated and a bigger dataset is currently being analysed has been included as part of the study limitations (pages 15-16 (lines 294-327)).

  • table 3 does not include patients with 0%.

We thank the reviewer for this valid point and have taken out the patients with 0% out of the table.

  • the ROC curves in figures 2a and 2b are not very specific

Thank you for this comment and we apologise for the confusion. Please could we check if the reviewer was referring to Figure 3, which we inadvertently placed underneath the Legend for Figure 2 (a-d)? The legend for Figure 3 has been placed on the next page.

  • the results and conclusions of the authors confirm what is already known in the literature

Thank you for this feedback and we appreciate that the reviewer may be referring to the safety of robotic surgery or other publications from our group; however, to our knowledge, this is the first dataset comparing P-POSSUM and ACS in the robotic cohort in gynaecological  oncology.

Reviewer 2 Report

Comments and Suggestions for Authors

Overall, I commend the authors on an excellent undertaking and manuscript.

A few minor suggestions/revisions to be addressed:

In a study like this, details RE: data capture are important.  How were post operative complications identified in your cohort? Do you capture readmission/presentation to other hospitals/jurisdictions?  

Why 153 records/patients?  Some justification/explanation should be documented

Looks like this population was quite well (low prevalence of comorbidities, relatively low BMI, low stage malignancy)

Please consider adding BMI ranges to Table 3 for clear reference (rather than just underweight, normal, overweight, obese class 1/2/3

Table 7- what statistical test was applied to assess your null hypothesis?  T test?  This should be noted

Conclusions:

Consider "suggests" not "confirms" in line 288- this was too small a study to claim 'confirmation' of superiority of one predictive model over another.

Comments on the Quality of English Language

Generally well written and organized.  Minor grammatical and organizational edits should be considered.  The full names (not just the acronyms) of the P-POSSUM and ACS-NSQIP should maybe appear in the abstract.

Author Response

Overall, I commend the authors on an excellent undertaking and manuscript. A few minor suggestions/revisions to be addressed:

In a study like this, details RE: data capture are important. How were post operative complications identified in your cohort? Do you capture readmission/presentation to other hospitals/jurisdictions?

We thank the reviewer for this helpful suggestion and have added the extra information to the methods section (lines 106-115, 123-131).

Why 153 records/patients? Some justification/explanation should be documented In our study the first 153 robotic cases for whom data were collected were analysed.

We have included a paragraph on study limitations stating that this is a pilot project with only 153 patients from a single site has been evaluated and a bigger dataset is currently being analysed (lines 106-111, 298-300).

Looks like this population was quite well (low prevalence of comorbidities, relatively low BMI, low stage malignancy)

We are very grateful to reviewer 2 for raising this point and have added these to limitations of our study as well as undertaking a multicentre analysis from other parts of the UK (lines

 301-308).

Please consider adding BMI ranges to Table 3 for clear reference (rather than just underweight, normal, overweight, obese class 1/2/3

Thank you for this comment. We have now added the BMI ranges to the table (Table 3).

Table 7- what statistical test was applied to assess your null hypothesis? T test? This should be noted

We have added the test used in the statistical analysis. (Lines 234-235)

Conclusions:

Consider "suggests" not "confirms" in line 288- this was too small a study to claim 'confirmation' of superiority of one predictive model over another.

We agree and thank the reviewer and have amended the article accordingly. (Line 330)

Reviewer 3 Report

Comments and Suggestions for Authors

Dear Authors

Thank you for a well written paper. The sample size for validating the risk prediction tools is extremely small and limits validity of your evaluation. There was no mortality in your dataset and so you cannot reliably validate the risk prediction tools for predicting this. I would recommend re-running your analysis with a larger dataset that includes deaths as occurred events

Author Response

Dear Authors

Thank you for a well written paper.

The sample size for validating the risk prediction tools is extremely small and limits validity of your evaluation.

We thank the reviewer for raising this valid point. We have included a paragraph on study limitations stating that this is a pilot project with only 153 patients from a single site has been evaluated and a bigger dataset is currently being analysed. (Lines 106-111, 298-

 300).

There was no mortality in your dataset and so you cannot reliably validate the risk prediction tools for predicting this. I would recommend re-running your analysis with a larger dataset that includes deaths as occurred events.

Thank you for highlighting this and while we absolutely agree that there is no mortality in our dataset, we think it is important to include the mortality analysis as it has shown null mortality rate prediction when ACS NSQIP was used, which the P-POSSUM model failed to do. Indeed a pilot cohort of 153 patients is insufficient to demonstrate mortality and an analysis on a larger cohort of patients is ongoing. The authors feel it would be helpful for clinicians to have the current research outcome available as it may aid choosing a more appropriate tool for prediction of postoperative complications.